# Targeting SHP2 with Natural Products: Exploring Saponin-Based Allosteric Inhibitors and Their Therapeutic Potential

**DOI:** 10.3390/cimb47050309

**Published:** 2025-04-27

**Authors:** Dong-Oh Moon

**Affiliations:** Department of Biology Education, Daegu University, 201, Daegudae-ro, Gyeongsan-si 38453, Gyeongsangbuk-do, Republic of Korea; domoon@daegu.ac.kr

**Keywords:** SHP2, inhibitor, natural products, saponins

## Abstract

SHP2, a non-receptor protein tyrosine phosphatase, plays a pivotal role in regulating intracellular signaling pathways, particularly the RAS/MAPK and PI3K/AKT cascades, which are critical for cellular proliferation, differentiation, and survival. Aberrant SHP2 activity, often driven by gain-of-function mutations, is implicated in oncogenesis and drug resistance, making it an attractive therapeutic target. Traditional inhibitors targeting SHP2’s catalytic site face limitations such as poor selectivity and low bioavailability. Recent advancements in allosteric inhibitors, specifically targeting SHP2’s tunnel site, offer improved specificity and pharmacokinetics. Natural products, especially saponins with their unique structural diversity, have emerged as promising candidates for SHP2 inhibition. This review explores the structural and functional dynamics of SHP2, highlights the potential of saponin-based inhibitors, and discusses their mechanisms of action, including their interactions with key residues in the tunnel site. The therapeutic potential of saponins is further emphasized by their ability to overcome the limitations of catalytic inhibitors and their applicability in combination therapies. Future directions include structural optimization to improve pharmacokinetics and the development of innovative strategies such as PROTACs to enhance the clinical utility of saponin-based SHP2 inhibitors.

## 1. Introduction

SHP2, a non-receptor protein tyrosine phosphatase encoded by the PTPN11 gene, is essential for intracellular signaling pathways [1,2]. It consists of two SH2 domains (N-SH2 and C-SH2) and a catalytic PTP domain [3,4]. In its autoinhibited state, the N-SH2 domain blocks the catalytic active site, preventing substrate access [5,6]. Upon activation by receptor tyrosine kinases (RTKs), SHP2 binds to phosphorylated tyrosine residues on signaling proteins, undergoing a conformational change that exposes its catalytic site [7]. This activation enables SHP2 to regulate key pathways such as RAS/MAPK, which controls cell proliferation and differentiation, and PI3K/AKT, which is crucial for cell survival and metabolism [8,9,10]. These roles make SHP2 vital for maintaining cellular homeostasis and supporting physiological functions.

In cancer, SHP2 is frequently overexpressed or hyperactivated due to gain-of-function (GOF) mutations, which destabilize its autoinhibited conformation and lead to constitutive activation [2,11]. Notable GOF mutations, such as E76D and D61G, amplify RAS/MAPK signaling, driving tumor proliferation, survival, and metastasis [12,13]. SHP2 hyperactivation has been implicated in several malignancies, including leukemia, breast cancer, and non-small cell lung cancer (NSCLC) [14]. In leukemia, SHP2 mutations contribute to juvenile myelomonocytic leukemia (JMML) and acute myeloid leukemia (AML), promoting uncontrolled proliferation [11]. Similarly, in breast cancer, SHP2 enhances HER2-driven signaling, facilitating resistance to targeted therapies [15]. In NSCLC, SHP2 integrates upstream signals from oncogenic RTKs, such as EGFR, and KRAS mutations, sustaining tumor growth and resistance to therapeutic interventions [16]. Beyond its catalytic function, SHP2 serves as a scaffolding protein, enabling interaction between RTKs and downstream effectors, further underscoring its pivotal role in oncogenesis and therapy resistance.

Efforts to target SHP2 initially focused on its highly conserved PTP catalytic site. However, this approach was hampered by significant challenges. The active site of SHP2 shares a high degree of structural similarity with other protein tyrosine phosphatases (PTPs), leading to poor selectivity and unintended off-target effects [17]. Additionally, the polar and solvent-exposed nature of the catalytic site limited the development of drug-like molecules with favorable pharmacokinetic properties, resulting in reduced clinical utility [4]. Consequently, active-site inhibitors failed to meet therapeutic expectations. The emergence of allosteric inhibitors marked a significant advancement in SHP2-targeted therapy [18,19,20]. Allosteric inhibitors bind to non-catalytic regions, such as the tunnel site formed at the interface of the N-SH2, C-SH2, and PTP domains. This binding stabilizes SHP2 in its autoinhibited conformation, simultaneously blocking its catalytic activity and scaffolding function. Unlike active-site inhibitors, allosteric inhibitors exhibit greater selectivity and improved pharmacokinetics, minimizing off-target effects. For example, SHP099, the first-generation allosteric inhibitor, demonstrated potent inhibition of SHP2 by stabilizing its inactive conformation and reducing RAS/MAPK pathway activity in preclinical models [7,21]. Second-generation inhibitors, such as TNO155 and RMC-4630, are currently undergoing clinical evaluation and have shown promise in disrupting SHP2-driven oncogenic signaling [22,23]. These inhibitors have also demonstrated potential in combination therapies by overcoming resistance to RTK inhibitors and modulating the immune microenvironment, enhancing T-cell activation and reducing immunosuppressive signals.

In recent years, several SHP2 inhibitors have advanced into clinical trials, reflecting the translational promise of SHP2-targeted therapy. Notable examples include SHP099, a first-in-class allosteric inhibitor; TNO155 (Novartis), which is currently in Phase I/II trials for advanced solid tumors; RMC-4630 (Revolution Medicines), which has demonstrated potent anti-tumor activity in RAS-driven cancers; PF-07284892 (Pfizer), also known as RLY-1971, which has entered clinical studies for combination regimens; and JAB-3312 (Jacobio Pharmaceuticals), which is undergoing trials in China and the U.S. (NCT03114319, NCT03634982, NCT03518554, and NCT03565003). These candidates highlight the diverse strategies being employed to target SHP2 in clinical settings, often showing nanomolar potency and favorable pharmacokinetics.

Natural products have emerged as an important resource for developing SHP2 inhibitors, offering a wide range of bioactive compounds with unique structural diversity [24,25]. Among these, saponins, a class of glycosylated compounds with triterpene or steroid backbones, have shown particular promise. Saponins are known for their broad pharmacological activities, including anti-inflammatory, anticancer, and immune-modulating effects, making them ideal candidates for cancer therapy. Notably, saponins have been shown to interact with various protein targets, including SHP2, through both catalytic and allosteric mechanisms. For example, polyphyllin D, a steroidal saponin, has demonstrated significant anti-cancer activity by allosterically inhibiting SHP2 with an IC50 of 15.3 µM [26]. This inhibition disrupts downstream pathways such as ERK/MAPK, reducing cancer cell viability and inducing apoptosis in leukemia cells. Such dual activity, i.e., the direct inhibition of SHP2 and the modulation of oncogenic pathways, positions saponins as promising candidates for SHP2-targeted drug discovery.

This review seeks to provide a comprehensive overview of SHP2’s structure, including its catalytic and allosteric inhibition sites and their relevance in therapeutic targeting. Special attention is given to natural products, with a particular focus on saponins, which exhibit unique structural features and potential as effective SHP2 inhibitors. By examining their mechanisms of action and interactions with SHP2, this review aims to highlight their significance in drug discovery and cancer therapy.

## 2. Structural and Functional Dynamics of SHP2

SHP2 is composed of three key domains: the N-terminal SH2 domain (N-SH2), the C-terminal SH2 domain (C-SH2), and the catalytic protein tyrosine phosphatase (PTP) domain, connected by flexible linkers and a C-terminal tail [2,27,28]. In its auto-inhibited state, SHP2 adopts a conformationally closed structure, where the N-SH2 domain occludes the PTP catalytic site, physically preventing substrate access and maintaining the enzyme in an inactive state [4,29]. This conformation is stabilized by intramolecular interactions between the N-SH2 and PTP domains.

Activation of SHP2 is triggered by binding phosphorylated tyrosine residues on RTKs or adaptor proteins to the SH2 domains [10,30]. These interactions disrupt the intramolecular blockade between the N-SH2 and PTP domains, exposing the catalytic site and transitioning SHP2 into an open, catalytically active conformation. This structural change enables SHP2 to perform its phosphatase function, which is critical in regulating cellular signaling pathways, particularly the RAS/MAPK pathway [31].

The phosphatase activity of SHP2 operates through a conserved PTP catalytic site composed of key residues such as Cys459, Arg465, Asp425, and Gln506, following a multi-step catalytic cycle [32,33,34,35]. The process begins with the stabilization of the substrate’s phosphotyrosine group within the active site, where Arg465 interacts with the phosphate group, aligning it with the catalytic Cys459, while Asp425 stabilizes the surrounding water molecules, preparing them for subsequent reactions. In the next step, Cys459 acts as a nucleophile, attacking the substrate’s phosphotyrosine group to form a covalent cysteinyl-phosphate intermediate, which results in the cleavage of the phosphate–tyrosine bond and the release of dephosphorylated tyrosine from the active site. This intermediate then undergoes hydrolysis, a process facilitated by a water molecule coordinated by Asp425 and Gln506, where the water molecule attacks the phosphate group, breaking the covalent bond and freeing the phosphate. Finally, the active site is regenerated to its original state as Cys459 is restored to its thiol form, allowing the catalytic cycle to reset and prepare for the next substrate. This precise sequence ensures the efficient dephosphorylation of substrates and highlights the critical roles of the conserved active site residues in SHP2’s enzymatic function.

SHP2 also contains an allosteric tunnel site formed by interactions between the N-SH2, C-SH2, and PTP domains. Key residues within the tunnel allosteric site include Thr108, Phe113, Arg111, Glu110, and Glu250 [19,36]. This site plays a pivotal role in stabilizing SHP2’s autoinhibited conformation when targeted by allosteric inhibitors. The structural organization and catalytic mechanism of SHP2 are illustrated in Figure 1.

## 3. Natural Products as Promising SHP2 Inhibitors

SHP2 plays a central role in the regulation of the Ras/MAPK signaling pathway, a critical cascade for cell proliferation, differentiation, and survival [37,38,39]. It functions downstream of RTKs such as EGFR, FGFR, and PDGFR, facilitating the activation of Ras, a key GTPase that transitions between an inactive GDP-bound state and an active GTP-bound state [28,30,37,39]. SHP2 enhances Ras activity by dephosphorylating inhibitory tyrosine residues, such as pY32 on Ras, which increases the stability of the GTP-bound active form [8,9]. Furthermore, SHP2 dephosphorylates GAPs (GTPase-activating proteins) such as RasA, thereby reducing their ability to inactivate Ras [9]. Additionally, SHP2 interacts with the GAB1-GRB2-SOS1 complex, promoting the guanine nucleotide exchange factor (GEF) activity of SOS1 to convert GDP-bound RAS into its GTP-bound form [40]. This sustained activation of RAS drives downstream signaling through the RAF/MEK/ERK phosphorylation cascade, connecting extracellular growth signals to nuclear transcription factors that regulate cellular processes such as proliferation and survival.

In the PI3K/AKT signaling pathway, SHP2’s role is context-dependent and influenced by the specific RTKs involved [9,10,41]. SHP2 can positively regulate this pathway by associating with free p85, the regulatory subunit of PI3K, thereby promoting AKT activation. This function is particularly evident in some cellular contexts where SHP2 also enhances ERK signaling. On the other hand, SHP2 can negatively regulate PI3K/AKT signaling by dephosphorylating the docking sites for GAB1 and PI3K, thus limiting their interaction. SHP2 also forms a complex with GAB2 and p85, further blocking PI3K activation [10,28,42,43,44,45]. This dual regulatory function highlights SHP2’s role as a finely tuned modulator of PI3K/AKT signaling, influencing processes such as metabolism, cell growth, and survival, depending on cellular and receptor-specific contexts.

SHP2’s critical roles in the RAS/MAPK and PI3K/AKT signaling pathways establish its significance as a therapeutic target, particularly in oncogenic contexts. Building on this foundation, natural products emerge as a promising avenue for SHP2 inhibition, offering diverse mechanisms of action and potential for targeted therapy.

Natural products offer significant advantages over synthetic compounds in drug discovery, providing diverse structural frameworks and extensive biological activity [46]. Historically, these compounds have played a critical role in addressing unknown diseases, demonstrating their efficacy through empirical use over centuries [47]. Their high structural diversity and biological versatility make natural products a valuable source for therapeutic development, either as direct drug candidates or scaffolds for novel drug design [48,49]. Recently, natural products have emerged as promising SHP2 inhibitors due to their multifaceted biological activities and ability to target SHP2 through a variety of mechanisms, ranging from active site inhibition to allosteric modulation. A detailed analysis of the natural products studied for SHP2 inhibition provides crucial insights into their binding mechanisms, interaction profiles, and therapeutic potential.

For example, terpenoids such as 3-acetoxylteuvincenone G (3-AG), cryptotanshinone (CTS), enoxolone, and celastrol primarily target the PTP active site of SHP2 [50,51,52]. Among these, celastrol demonstrates the strongest inhibitory effect, with an IC50 of 3.3 ± 0.6 µM, by interacting with critical residues such as Arg465, Gln510, and Lys366. These interactions stabilize the catalytic domain and block substrate processing through hydrogen bonds and hydrophobic interactions, ensuring high specificity and efficiency. Celastrol, in particular, has been shown to inhibit downstream RAS/MAPK signaling, reducing cell proliferation and inducing apoptosis in cancer cells, such as lung and liver carcinoma models [53].

Fumosorinone is a 2-pyridone alkaloid that acts as a selective non-competitive inhibitor of SHP2 by binding to its PTP loop, with an IC50 of 6.31 µM. It inhibits SHP2-dependent Ras/ERK signaling downstream of EGFR and prevents tumor cell invasion by downregulating Src signaling and phosphorylation of key molecules such as paxillin and Src [54].

Polyphenols, exemplified by ellagic acid, exhibit exceptional potency, with an IC50 of 0.69 ± 0.07 µM, the lowest among the natural products studied [55]. Ellagic acid forms hydrogen bonds with residues like Trp423, Gly427, and Gln506 in the PTP active site. This compound effectively inhibits the SHP2-mediated oncogenic signaling pathways, such as ERK and AKT, leading to reduced tumor growth and apoptosis in breast and colon cancer cells.

Polyketides such as tautomycetin (TTN) inhibit SHP2 competitively, with an IC50 of 2.9 ± 0.2 µM [44]. TTN binds to Arg465, Ser460, and Gln510, mimicking the natural substrate’s binding mode. This interaction blocks downstream signaling and sensitizes cancer cells to apoptosis-inducing agents, highlighting its potential for combination therapies [56].

Flavonoids, including quercetin, provide moderate SHP2 inhibition, with an IC50 of 10.17 ± 0.21 µM [57]. Quercetin interacts with Arg465, Gln506, and Trp423 in the active site, disrupting the SHP2-mediated activation of the JAK/STAT pathways. This results in decreased cell viability and increased apoptosis, particularly in hepatocellular carcinoma cells when combined with interferon-α.

Xanthones, such as phomoxanthone A (PXA) and phomoxanthone B (PXB), act as non-competitive inhibitors, with IC50 values of 20.47 ± 4.45 µM and 11.86 ± 3.02 µM, respectively [58]. These compounds stabilize SHP2 in its inactive conformation by interacting with residues Lys366, Asp425, and Arg465. Their inhibition of SHP2 disrupts cancer-related signaling pathways and induces apoptosis in leukemia cell lines.

β-lactam antibiotics like cefsulodin selectively inhibit SHP2 with an IC50 of 16.8 ± 2.0 µM [59]. Cefsulodin targets Ser460, Gly464, and Arg465 within the active site. Its sulfophenyl acetic amide (SPAA) moiety mimics the phosphotyrosine structure, effectively blocking catalytic activity. This inhibition reduces oncogenic signaling and sensitizes cells to chemotherapeutic agents.

Saponins, represented by polyphyllin D, target the tunnel allosteric site with an IC50 of 15.3 ± 0.8 µM [26]. Polyphyllin D binds to the residues Arg111, Phe113, and His114, stabilizing SHP2’s autoinhibited conformation. This prevents its activation, suppressing downstream RAS/MAPK pathways, inducing apoptosis, and reducing proliferation in leukemia and lung cancer cells [24].

The integrated analysis of natural products targeting SHP2 underscores their diverse inhibitory mechanisms, including competitive and non-competitive binding, as well as active and allosteric site interactions. The key residues involved, such as Arg465, Gln506, and Ser460, are consistently engaged across various compound classes, highlighting their critical role in SHP2 inhibition. The centrality of SHP2 in regulating key signaling pathways, such as RAS/MAPK and PI3K/AKT, highlights its critical role in cellular homeostasis and oncogenesis. The exploration of natural products as SHP2 inhibitors not only underscores their structural diversity and biological activity but also demonstrates their ability to effectively target both active and allosteric sites. These findings reinforce the therapeutic value of natural products and set the stage for further studies to refine their potency and applicability in clinical settings. These results are summarized in Table 1 and Figure 2 for clarity and reference.

While docking studies have revealed the binding sites of these natural products, several others have been reported as SHP2 inhibitors, despite the lack of detailed binding site information. Postinin A and postinin B, sesquiterpenes, exhibit SHP2 inhibition with IC50 values of 3.9 mg/mL and 5.3 mg/mL, respectively, although their exact mechanisms remain unclear [60]. Among the triterpenoids, ursolic acid inhibits SHP2 at the catalytic site with IC50 values between 2.73 µM and 3.33 µM, and its derivative, UA0713, demonstrates enhanced potency with an IC50 of 0.76 µM [61]. Additionally, 3S,23R-dihydroxycycloart-24-en-26-oic acid, another triterpenoid, inhibits SHP2-related signaling pathways with an IC50 of 19 µM [62]. Arthrianhydride A and B, from the anhydride class, inhibit SHP2, with arthrianhydride B showing a higher inhibition rate of 63.02% compared to 12.80% for arthrianhydride A [63]. Deflectin 1c and deflectin 2b, azaphilones, are potent SHP2 inhibitors with IC50 values of 0.8 µM and 0.7 µM, respectively, likely targeting active or allosteric sites [64]. The lignan suchilactone inhibits SHP2 activity by targeting its active site, suppressing downstream signaling pathways such as p-ERK and p-AKT, promoting apoptosis, and reducing proliferation [65]. Lastly, HLP46, a sterol, significantly reduces p-SHP2 levels, impacting cell migration and invasion [66]. These findings highlight the diversity of natural products as SHP2 inhibitors, providing a foundation for future research and optimization to enhance potency and selectivity.

## 4. Exploring Saponins as Allosteric Inhibitors of SHP2

Current research on natural products targeting SHP2 has identified polyphyllin D as the only reported allosteric inhibitor specifically targeting the tunnel site of SHP2 [26]. This observation underscores the need for further investigation into whether other saponin-based compounds, including polyphyllin D, can effectively inhibit the SHP2 tunnel allosteric site. Allosteric inhibitors targeting this site offer significant advantages over catalytic site inhibitors, including enhanced selectivity, reduced off-target effects, and improved pharmacokinetic properties. These benefits highlight the potential of tunnel site inhibitors as a therapeutic strategy for addressing limitations associated with catalytic site inhibitors.

To explore this potential, 29 saponin compounds were selected for their structural diversity and bioactivities relevant to cancer therapy [67,68,69,70]. Saponins are glycosylated compounds characterized by a triterpenoid or steroidal aglycone backbone linked to sugar moieties, enabling diverse interactions with protein targets [71,72]. The compounds represent various saponin subgroups, including polyphyllin, diosgenin derivatives, ruscogenin, solasodine derivatives, yamogenin, sarsasapogenin, and cholestane saponins [73]. These compounds were selected with the aim of including a wide range of chemical scaffolds and functional groups, facilitating a comprehensive evaluation of SHP2 inhibition. All selected compounds, including polyphyllin D (CID: 11018329), polyphyllin B (CID: 328441), diosgenin (CID: 119245), and ophiopogonin D (CID: 46173859), were retrieved from the PubChem database (https://pubchem.ncbi.nlm.nih.gov/), ensuring publicly accessible chemical and structural data for this study (Figure 3; accessed on 1 March 2025).

The potential of these 29 saponin compounds as SHP2 inhibitors was assessed through molecular docking using the CB-Dock2 platform. SHP2’s structure (PDB ID: 5EHR) was prepared for docking by removing the water molecules, optimizing it for interaction studies. CB-Dock2, an advanced blind docking tool, automatically scans protein surfaces to identify potential binding cavities and predicts the most favorable ligand poses based on cavity volume, shape, and physicochemical properties [74,75]. This approach ensures an unbiased evaluation of ligand binding potential, particularly for the SHP2 tunnel allosteric site, a critical regulatory region stabilizing SHP2’s autoinhibited conformation.

Docking simulations provided binding energies, predicted binding poses, and indicated interaction residues for each compound. Hydrogen bonds, hydrophobic contacts, and van der Waals interactions were analyzed to determine each compound’s inhibitory potential. Visualization of the protein–ligand interactions was conducted using Discovery Studio 2024 Client to ensure the precise analysis and communication of docking results.

As summarized in Table 2, the docking scores for the saponin compounds ranged from −7.8 kcal/mol to −11.5 kcal/mol, with polyphyllin II exhibiting the strongest binding affinity (−11.5 kcal/mol). This high binding potential reflects polyphyllin II’s robust interactions within the SHP2 tunnel allosteric site, including hydrogen bonds with Glu249, Glu250, and Thr218 and hydrophobic contacts with Leu125 and Leu233. These interactions highlight its ability to stabilize SHP2’s autoinhibited conformation and effectively inhibit its activity.

Ophiopogonin A demonstrated a similarly high binding affinity (−11.0 kcal/mol), forming hydrogen bonds with Thr218, Glu250, and Leu236. The compound’s hydroxyl-rich regions facilitated polar interactions, while its hydrophobic core complemented nonpolar tunnel site regions. Polyphyllin B, with a docking score of −10.8 kcal/mol, formed hydrogen bonds with Arg111, Glu249, and Thr253, along with hydrophobic contacts involving Leu125, Gly246, and Glu250. The structural features of these compounds, such as glycosylated regions and hydrophobic backbones, contributed significantly to their selective targeting of the SHP2 tunnel allosteric site. The docking results for polyphyllin II, ophiopogonin A, and polyphyllin B, as analyzed using Discovery Studio 2024 Client, are shown in Figure 4.

The docking results underscore the ability of saponin compounds to selectively target SHP2’s tunnel allosteric site, a critical regulatory region that offers distinct advantages over catalytic site inhibition. This study expands our understanding of SHP2-targeting natural products and highlights the potential of saponins as allosteric inhibitors capable of stabilizing SHP2’s inactive conformation. These findings provide a strong foundation for the experimental validation and further optimization of these lead compounds.

## 5. Future Strategies for SHP2 Inhibition Using Saponin-Based Natural Products

Traditional SHP2 inhibitors targeting the catalytic site have faced significant challenges, including poor selectivity due to the conserved nature of the active site among protein tyrosine phosphatases (PTPs) and low bioavailability, highlighting the need for innovative therapeutic strategies [76].

Saponins, characterized by their unique structural composition of hydrophobic aglycones and hydrophilic sugar moieties, present a novel approach for SHP2 inhibition. These compounds possess diverse chemical scaffolds that enable interactions with SHP2’s tunnel allosteric site, a critical regulatory region that stabilizes SHP2 in its autoinhibited conformation. This review highlights the promising potential of saponin-based inhibitors such as polyphyllin D, polyphyllin II, and ophiopogonin A, which effectively bind to the tunnel site, targeting key residues like Arg111, Glu250, and Phe113.

### 5.1. Combination Strategies for Enhanced Therapeutic Outcomes

Saponins with strong inhibitory effects on SHP2’s tunnel site, such as polyphyllin II, could be combined with inhibitors targeting complementary pathways, such as MEK or PI3K inhibitors, to address adaptive resistance mechanisms in tumors. This dual inhibition strategy could disrupt redundant signaling pathways, enhancing the therapeutic response and delaying the onset of resistance.

### 5.2. Enhancing Immune-Mediated Cancer Clearance

Incorporating saponin-based SHP2 inhibitors into combination regimens using immune checkpoint inhibitors (e.g., anti-PD-1/PD-L1 antibodies) offers a compelling approach to leverage SHP2’s role in the tumor microenvironment. SHP2’s suppression of T-cell activation through PD-1-mediated signaling can be counteracted, enhancing immune system activity against tumors and promoting durable anti-tumor responses [77,78].

### 5.3. Optimizing Saponin-Based Therapeutics

To overcome pharmacokinetic limitations and hemolytic toxicity, multiple strategies for modifying saponins have been proposed. Structural simplification through the selective removal of sugar moieties can reduce molecular weight and improve membrane permeability. Additionally, SAR studies have shown that modifying the aglycone backbone (e.g., triterpenoid vs. steroidal core) and side-chain substitutions (e.g., hydroxyl or acetyl groups) can affect both potency and cytotoxicity. Prodrug strategies, such as esterification to enhance lipophilicity and absorption, and conjugation with polymeric carriers or nanoparticles, have also been explored to enhance bioavailability and tumor-specific delivery. Recent advances in semisynthetic analogs—such as ginsenoside CK and derivatives of oleanolic acid—demonstrate that rational modifications can maintain bioactivity while reducing hemolysis. Such structure–activity relationship (SAR) approaches are essential for translating saponin scaffolds into clinically viable SHP2 inhibitors.

### 5.4. Enhancing Binding Affinity

Structural refinement of saponins to increase hydrogen bonding and hydrophobic interactions with key tunnel site residues, including Arg111 and Glu250, could further improve their binding affinity and inhibitory efficacy. Computational modeling and structure-based drug design approaches could guide these optimizations.

### 5.5. Developing PROTAC-Based Therapies

Saponin compounds with a high affinity for the tunnel allosteric site hold potential as scaffolds for developing proteolysis-targeting chimeras (PROTACs). These PROTACs could facilitate the targeted degradation of SHP2, offering a strategy for its irreversible inactivation in cancer therapy [79,80]. This approach offers a mechanism to irreversibly deactivate SHP2, particularly in cancers driven by SHP2 hyperactivation, potentially providing more durable therapeutic effects.

## 6. Conclusions

This review highlights the potential of saponin-based compounds as promising SHP2 inhibitors, with a particular focus on their ability to target the tunnel allosteric site. Compounds such as polyphyllin D, polyphyllin II, and ophiopogonin A have demonstrated significant inhibitory potential by stabilizing SHP2 in its autoinhibited conformation. The unique structural features of saponins allow for diverse interactions with key residues in the tunnel site, such as Arg111, Glu250, and His114. These interactions underline their potential for the selective inhibition of SHP2, making them attractive candidates for further therapeutic exploration.

Although SHP2 inhibitors currently in clinical development demonstrate high potency and favorable drug-like properties, exploring alternative scaffolds such as saponins remains valuable. Saponin-based compounds offer unique structural frameworks that may provide novel modes of SHP2 inhibition, especially via tunnel allosteric binding, which could lead to subtype-selective inhibition or complementary effects in resistant tumors. Additionally, their natural origin and inherent bioactivities—such as anti-inflammatory, immunomodulatory, and pro-apoptotic effects—make saponins attractive as multitarget agents or adjuvants in combination therapies. The pursuit of saponin scaffolds also aligns with natural product-inspired drug discovery, which has historically led to structurally unique and biologically effective therapeutic agents.

However, saponins also present certain structural limitations that may hinder their clinical application. Their large molecular size and complex glycosylation patterns can result in poor membrane permeability, limiting their bioavailability [81]. Additionally, the hydrophilic nature of the sugar moieties can lead to rapid clearance and low metabolic stability, further reducing their effectiveness as drug candidates. These drawbacks highlight the need for structural optimization to improve pharmacokinetic properties while maintaining high binding affinity and specificity.

While saponins exhibit promising inhibitory activity against SHP2, a critical limitation to their clinical application is their well-documented hemolytic effect, which arises from their amphiphilic structure that disrupts cholesterol-containing membranes, such as those in erythrocytes. This hemolysis poses significant safety concerns and can severely limit the therapeutic window of saponin-based drug candidates. To address this, various strategies have been explored to mitigate the hemolytic activity of saponins. One promising approach involves structural modification through semisynthetic or synthetic derivatization to reduce membrane-disruptive interactions, while preserving pharmacological activity. For instance, glycosidic moiety modifications and acylation patterns have been optimized in ginsenosides to attenuate hemolysis without compromising bioactivity. In the context of SHP2 inhibition, although no studies have directly reported non-hemolytic saponin derivatives targeting SHP2, several analogs of ginsenosides and oleanane-type saponins have shown reduced hemolytic profiles, alongside preserved anticancer activities. These efforts underscore the potential for rational design to decouple hemolytic toxicity from SHP2-targeted efficacy, making structural optimization a pivotal step in the development of clinically viable saponin-based SHP2 inhibitors

Future research should focus on modifying saponin structures to overcome these challenges. Strategies such as reducing glycosylation, enhancing lipophilicity, or incorporating prodrug approaches could improve their pharmacokinetic profiles. Furthermore, combining saponin-based SHP2 inhibitors with other therapeutic agents, such as immune checkpoint inhibitors or kinase inhibitors, may provide synergistic effects and improve clinical outcomes. As these limitations are addressed, saponins exhibit the potential to serve as a foundation for the development of novel, effective SHP2-targeted therapies.

## Figures and Tables

**Figure 1 cimb-47-00309-f001:**
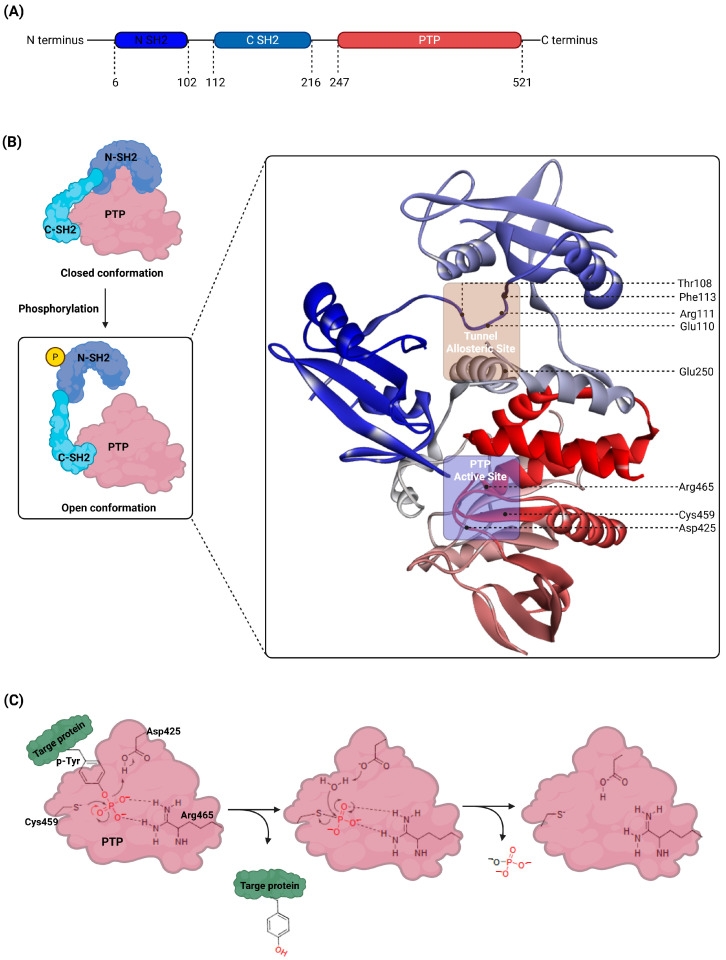
Structural organization and catalytic mechanism of SHP2. (**A**) Schematic representation of SHP2’s domain structure, including the N-terminal SH2 domain (N-SH2), the C-terminal SH2 domain (C-SH2), and the protein tyrosine phosphatase (PTP) domain, connected by flexible linkers and extending to the C-terminal tail. (**B**) Structural transitions of SHP2 between the closed (autoinhibited) and open (catalytically active) conformations. The 3D structure of SHP2, shown within the boxed region, was obtained from the Protein Data Bank (PDB ID: 5EHR). The tunnel allosteric sites (Thr108, Phe113, Arg111, Glu110, and Glu250) and the PTP active sites (Cys459, Arg465, Asp425, and Gln506) are highlighted. (**C**) Detailed depiction of SHP2’s catalytic mechanism at the PTP active site, including substrate stabilization, cysteinyl-phosphate intermediate formation, hydrolysis, and active site regeneration. Key residues involved in the reaction (Cys459, Arg465, Asp425) are shown.

**Figure 2 cimb-47-00309-f002:**
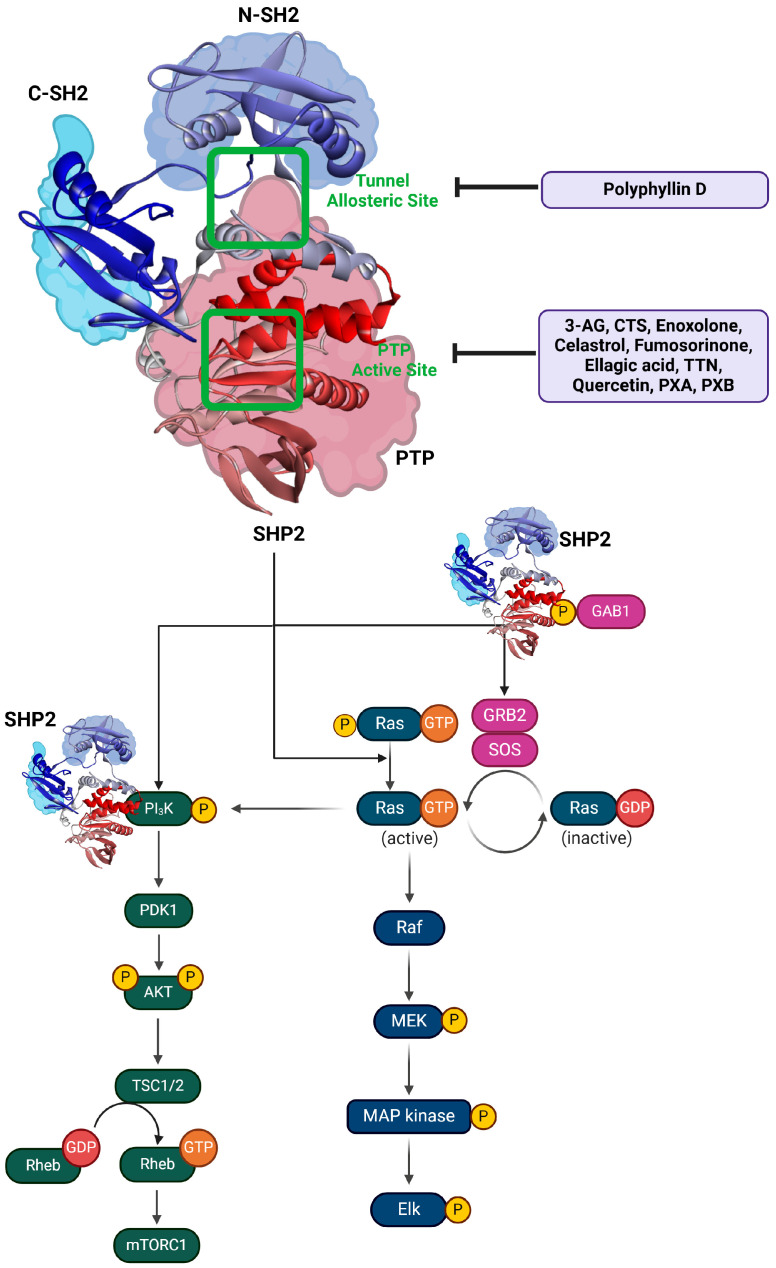
SHP2 structure and natural compounds targeting RAS/MAPK and PI3K/AKT pathways. This figure highlights the structural domains of SHP2, including the N-SH2, C-SH2, and PTP domains, focusing on the tunnel allosteric site, targeted by polyphyllin D, and the PTP active site, inhibited by natural compounds such as 3-AG, CTS, and celastrol. SHP2 regulates key signaling pathways, including RAS/MAPK and PI3K/AKT, critical for cell proliferation and survival. In the RAS/MAPK pathway, SHP2 enhances RAS activation by dephosphorylating inhibitory tyrosine residues and interacting with the GAB1-GRB2-SOS complex to promote nucleotide exchange. This activates downstream kinases like RAF, MEK, and ERK, leading to transcription factor regulation and cellular growth. In the PI3K/AKT pathway, SHP2 plays dual roles, either activating AKT via p85-PI3K association or suppressing the pathway by dephosphorylating GAB1 and PI3K docking sites. These interactions modulate cellular metabolism, growth, and survival. Natural compounds targeting SHP2 influence these pathways, offering therapeutic potential for diseases with dysregulated signaling.

**Figure 3 cimb-47-00309-f003:**
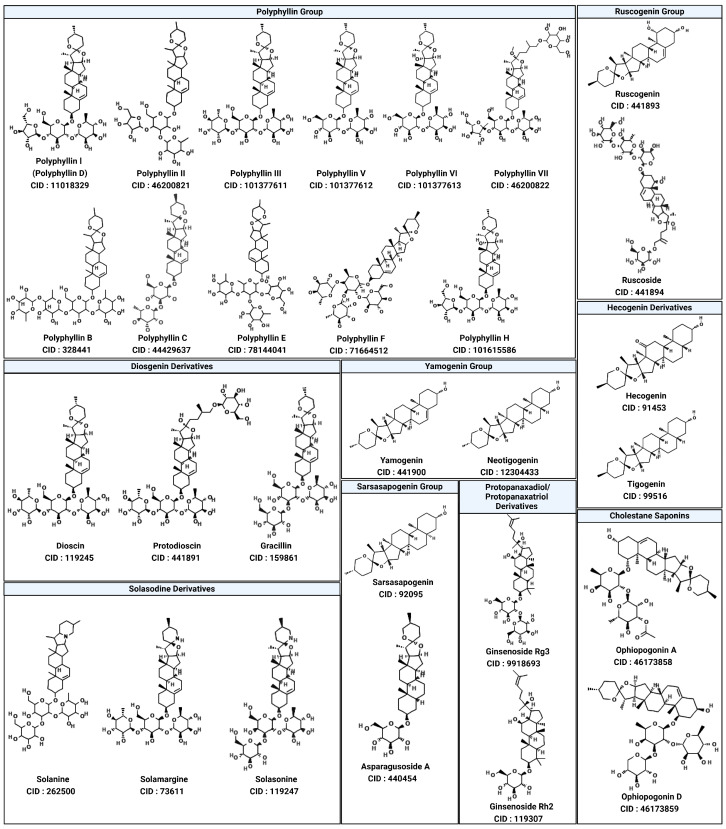
Structural diversity of selected saponin compounds. This figure illustrates the chemical structures and PubChem compound identifiers (CIDs) of the saponins selected for this study, grouped by their structural categories. The compounds represent diverse subclasses, including the polyphyllin group, diosgenin derivatives, ruscogenin group, solasodine derivatives, yamogenin group, sarsasapogenin group, and cholestane saponins. Each compound’s structural variations regarding the backbone and sugar moieties contribute to their potential interactions with SHP2. The structural diversity of these compounds underpins their selection for evaluating SHP2 inhibition and understanding structure–activity relationships.

**Figure 4 cimb-47-00309-f004:**
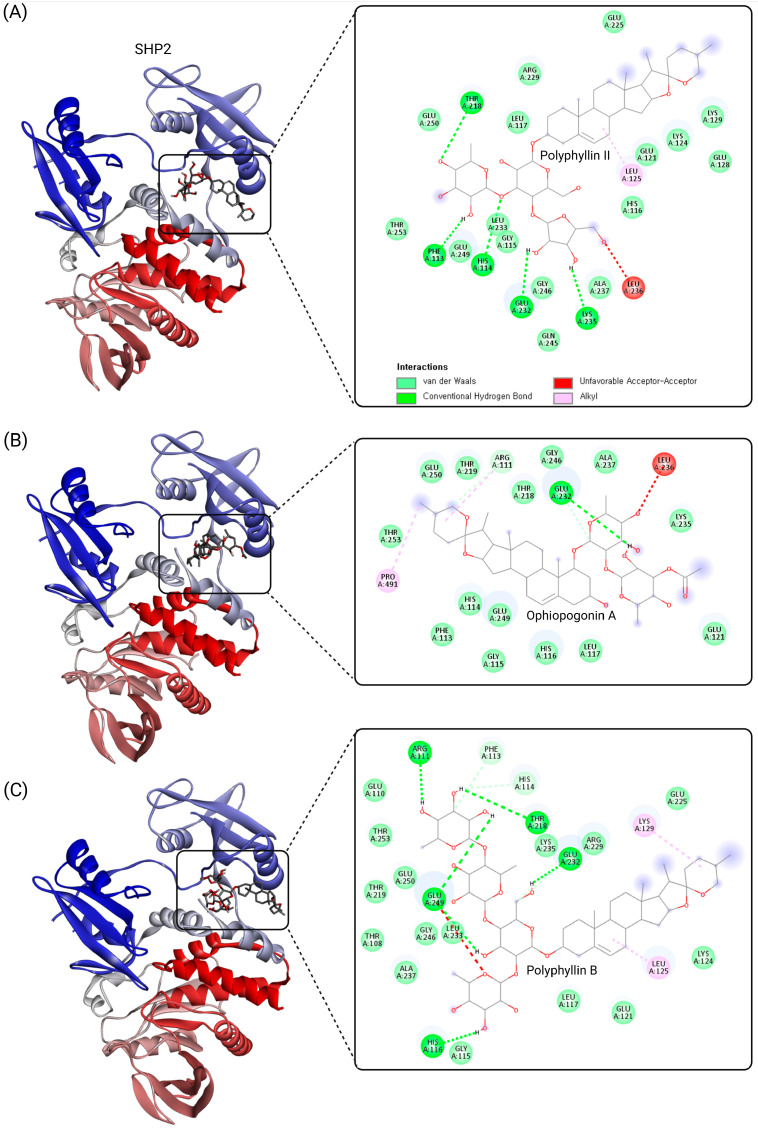
Visualization of ligand–protein interactions at the SHP2 tunnel allosteric site for polyphyllin II, ophiopogonin A, and polyphyllin B. The figure illustrates the molecular docking results for three selected saponin compounds, polyphyllin II (**A**), ophiopogonin A (**B**), and polyphyllin B (**C**), within the SHP2 tunnel allosteric site. The docking was performed using CB-Dock2, and the ligand–protein complexes were visualized using Discovery Studio 2024 Client. Key interactions, including conventional hydrogen bonds, van der Waals forces, alkyl interactions, and unfavorable acceptor–acceptor contacts, are highlighted. The SHP2 structure (PDB ID: 5EHR) is shown in cartoon representation, with the tunnel site residues labeled. These visualizations provide insights into the binding modes and interaction profiles of saponin compounds, emphasizing their potential as SHP2 allosteric inhibitors.

**Table 1 cimb-47-00309-t001:** Natural products targeting SHP2 and their mechanisms of inhibition. This table summarizes the natural products reported as SHP2 inhibitors, highlighting their IC50 values, inhibitory sites, and key interacting residues.

Class	Compound	IC50 (µM)	Inhibitory Site	Contact Residues	Ref.
Terpenoids	3-acetoxylteuvincenone G(3-AG) 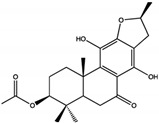	10.79 ± 0.14	PTP Active Site	Lys260, Lys492, Gln256, Gln257, Cys259, Tyr263, Glu313, Gln495, Met496, Arg498, Ser499	[50]
Cryptotanshinone (CTS) 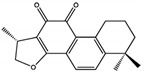	22.50	PTP Active Site	Lys358, Arg362, Lys364, Ser 365	[51]
Enoxolone 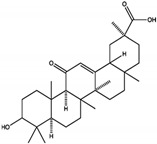	9.6 ± 3.2	PTP Active Site	Arg465, Gln510	[52]
Celastrol 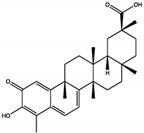	3.3 ± 0.6	PTP Active Site	Arg465, Gln510	[52]
Alkaloids	Fumosorinone 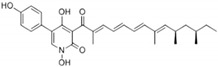	6.31	PTP Active Site (Non-Competitive)	Cys459, Gly427, Ala461, Ser460, Arg465, Tyr279, Lys366	[54]
Polyphenols	Ellagic acid 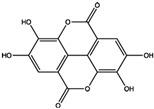	0.69 ± 0.07	PTP Active Site	Trp423, Gly427, Ser460, Gln506, Gln510	[55]
Polyketide	Tautomycetin (TTN) 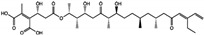	2.9 ± 0.2	PTP Active Site	Arg465, Ala461, Ser460, Gln510, Lys366	[56]
Flavonoids	Quercetin 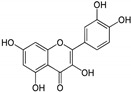	10.17 ± 0.21	PTP Active Site	Arg362, Trp423, Asp425, Arg465, Gln506, Gln510	[57]
Xanthones	Phomoxanthone A (PXA) 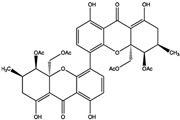	20.47 ± 4.45	PTP Active Site (Non-Competitive)	Lys366, Asp425, Gly427, Ser460, Gly464, Arg465, Gln506, Gln510	[58]
Phomoxanthone B (PXB) 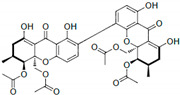	11.86 ± 3.02	PTP Active Site (Non-Competitive)	Lys366, Asp425, Gly427, Ser460, Gly464, Arg465, Gln506, Gln510	[58]
β-Lactam Antibiotic	Cefsulodin 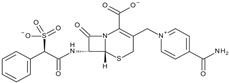	16.8 ± 2.0	PTP Active Site	Ser460, Ala461, Ile463, Gly464, Arg465, Lys366, Tyr279, Gln506	[59]
Saponins	Polyphyllin D 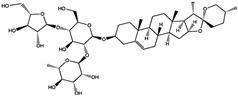	15.3 ± 0.8	Tunnel Allosteric Site	Arg111, Phe113, His114	[26]

**Table 2 cimb-47-00309-t002:** Docking results for saponin compounds with SHP2. The table includes the docking scores (kcal/mol) and key interacting residues identified during the docking of 29 saponin compounds with the SHP2 tunnel allosteric site.

Compounds	Binding Site	Vina Score (kcal/mol)	Contact Residues
SHP099 (positive control) ** 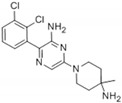 **	Tunnel site	−10.4	Conventional Hydrogen Bond: Thr218, Arg229, Glu249, Thr253, His114, Arg111, Glu110 Van der Waals Interactions: Leu125, Glu232, Glu225, Lys129, Thr219, Glu250 Alkyl Interaction: Lys129
Polyphyllin I (Polyphyllin D)	Tunnel site	−9.6	Conventional Hydrogen Bond: Thr218, Arg229, Glu249, Thr253, His114, Arg111, Glu110 Van der Waals Interactions: Phe113, Leu125, Glu232, Glu225, Lys129, Thr219, Glu250 Alkyl Interaction: Lys129
Polyphyllin II	Tunnel site	−11.5	Conventional Hydrogen Bond: Thr 218, Arg 229, Glu 249, Thr 253, His 114 Van der Waals Interactions: Leu 125, Glu 232, Glu 225, Lys 129, Thr 219, Glu 250 Alkyl Interaction: Leu 125 Unfavorable Acceptor–Acceptor: Leu 236
Polyphyllin III	Tunnel site	−10.1	Conventional Hydrogen Bond: Thr 218, Glu 249, Thr 219, His 114 Van der Waals Interactions: Leu 233, Glu 225, Lys 129, Leu 125, Gly 115 Alkyl Interaction: Lys 129 Unfavorable Donor–Donor: Arg 229
Polyphyllin V	Tunnel site	−7.8	Conventional Hydrogen Bond: Arg 220, Asn 217, Arg 111 Van der Waals Interactions: Ser 499, Tyr 263, Phe 314, Met 496, Gln 495 Unfavorable Donor–Donor: Lys 131
Polyphyllin VI	Tunnel site	−8.7	Conventional Hydrogen Bond: Thr 218, Arg 229, Glu 249, Thr 253, His 114, Arg 111, Glu 121, Ala 237, Leu 236 Van der Waals Interactions: Leu 233, Glu 250, Gly 246, Ser 118, His 116 Alkyl Interaction: Arg 111
Polyphyllin VII (Polyphyllin G)	Tunnel site	−8.9	Conventional Hydrogen Bond: His114, Arg229, Glu249, Glu128, Thr218 Carbon Hydrogen Bond: Gly115, Ala237 Van der Waals Interactions: Arg231, Glu121, Leu117, Leu125, Leu233, Leu236, Gly246, Glu250 Unfavorable Interactions: Lys124 (Donor–Donor), Ala237 (Acceptor–Acceptor)
Polyphyllin B	Tunnel site	−10.8	Conventional Hydrogen Bond: Arg111, Glu110, Glu249, Thr218, Thr253 Carbon Hydrogen Bond: Ala237 Van der Waals Interactions: His114, Gly115, Leu233, Gly246, Glu250 Pi-Donor Hydrogen Bond: His116
Polyphyllin C	Tunnel site	−9.0	Conventional Hydrogen Bond: Arg229, His114, Glu249 Carbon Hydrogen Bond: Gly115, Gly246 Van der Waals Interactions: Glu225, Glu232, His116, Phe113, Thr108, Glu139
Polyphyllin E	Tunnel site	−8.9	Conventional Hydrogen Bond: His116, Glu121, Arg229, Glu232, Glu249 Carbon Hydrogen Bond: Ala237 Van der Waals Interactions: Leu117, Lys124, Glu128, Glu139, Leu233, Leu236, Lys235, Gly246 Pi-Alkyl Interaction: Leu117
Polyphyllin F	Tunnel site	−8.1	Conventional Hydrogen Bond: Asn 336, Asn 339, Glu 299, Asp 296, Asp 340 Van der Waals Interactions: Ser 302, Gly 295, Val 301, Val 368, Val 382 Alkyl Interaction: Lys 369 Unfavorable Donor–Donor: Arg 343
Polyphyllin H	Tunnel site	−9.2	Conventional Hydrogen Bond: Arg 111, Thr 218, Arg 229, Thr 253, Glu 249, His 114, Glu 232 Van der Waals Interactions: Pro 491, Leu 125, Leu 117, Glu 110, Phe 113, Gly 246
Dioscin	Tunnel site	−9.2	Conventional Hydrogen Bond: Glu249, Glu250, Thr218, Thr219 Carbon Hydrogen Bond: Gly115, Ala237 Van der Waals Interactions: His114, His116, Leu117, Leu125, Leu233, Leu236, Gly246 Unfavorable Interaction: Arg229 (Donor–Donor)
Protodioscin	Tunnel site	−9.4	Conventional Hydrogen Bond: Glu249, Glu250, Thr218, Thr253, Arg111, Glu110, Thr108, Phe113 Van der Waals Interactions: Gly115, His116, Leu117, Arg220, Leu125, Glu121, Ala237 Unfavorable Interaction: Arg229 (Donor–Donor)
Gracillin	Tunnel site	−8.1	Conventional Hydrogen Bond: Asn336, Asn339, Asp340, Asp296 Van der Waals Interactions: Tyr380, Gly381, Val301, Val382, Pro300
Ruscogenin	Tunnel site	−9.3	Van der Waals Interactions: Glu121, Glu232, Glu249, His114, Thr218, Thr219, Glu250 Alkyl Interaction: Leu117, Leu125
Ruscoside	Tunnel site	−8.8	Conventional Hydrogen Bond: Arg111, Arg229, Arg220, Glu110, Glu249, Glu250, Lys124, Lys129, Asn222, Asp487 Van der Waals Interactions: Phe113, Trp112, Thr108, Ser109, Thr253 Unfavorable Interaction: Asp487 (Donor–Donor)
Hecogenin	Tunnel site	−8.9	Conventional Hydrogen Bond: Thr218, Glu250 Van der Waals Interactions: His114, Glu249, Thr219, Thr253, Gly246, Leu233 Alkyl Interaction: Leu125
Tigogenin	Tunnel site	−8.9	Conventional Hydrogen Bond: Thr218, Glu121 Van der Waals Interactions: Thr219, Thr253, Glu249, Glu250, His114, His116 Alkyl Interaction: Leu117
Solanine	Tunnel site	−8.4	Conventional Hydrogen Bond: Thr218, Thr219, Glu249, Ala237 Van der Waals Interactions: His116, Glu250, Thr253, Gly115, Gly246, Leu254 Alkyl Interaction: Arg111, Pro491 Unfavorable Interaction: His114 (Donor–Donor)
Solamargine	Tunnel site	−9.5	Conventional Hydrogen Bond: Glu232, Glu249, Thr218, Thr219 Van der Waals Interactions: His114, His116, Glu250, Gly246, Leu233, Ala237 Alkyl Interaction: Leu125, Lys124, Lys129
Solasonine	Tunnel site	−9.0	Conventional Hydrogen Bond: Glu232, Thr218, Ser228, Arg284 Van der Waals Interactions: His114, Glu249, Thr219, Thr253, Leu117, Leu233 Alkyl Interaction: Arg111 Unfavorable Interaction: Arg284 (Donor–Donor)
Yamogenin	Tunnel site	−9.3	Van der Waals Interactions: Leu125, Leu233, Glu249, Glu250, Thr218, Thr219 Pi-Alkyl Interaction: Leu117, His114
Neotigogenin	Tunnel site	−8.9	Conventional Hydrogen Bond: Glu121, Thr218 Van der Waals Interactions: Thr219, Thr253, Glu249, Glu250 Pi-Alkyl Interaction: Leu117 Alkyl Interaction: Arg111
Ophiopogonin A	Tunnel site	−11.0	Conventional Hydrogen Bond: Glu232, Thr218 Van der Waals Interactions: Gly246, Glu249, Glu250, Thr253, His116, Ala237 Pi-Alkyl Interaction: Pro491 Unfavorable Interaction: Leu236 (Acceptor–Acceptor)
Ophiopogonin D	Tunnel site	−10.1	Conventional Hydrogen Bond: Glu121, His114, Thr218 Van der Waals Interactions: Ala237, Leu236, Leu233, Glu249, Glu250, Thr219 Alkyl Interaction: Arg111 Carbon Hydrogen Bond: His116
Sarsasapogenin	Tunnelsite	−8.9	Conventional Hydrogen Bond: Thr218 Van der Waals Interactions: Thr219, Thr253, Glu249, Glu250, His114 Alkyl Interaction: Leu117, Arg111 Carbon Hydrogen Bond: His116
Asparagusoside A	Tunnel site	−8.4	Conventional Hydrogen Bond: Glu225, Arg229 Van der Waals Interactions: Glu232, Glu249, Gly246, Ala237 Alkyl Interaction: Leu117, Leu125 Unfavorable Interaction: Leu236 (Acceptor–Acceptor)
Ginsenoside Rg3	Tunnel site	−8.8	Conventional Hydrogen Bond: Glu121, Thr218, Arg229 Van der Waals Interactions: Leu125, Leu236, Glu249, Thr219, Ala237 Alkyl Interaction: Arg111, Pro491, Leu254 Unfavorable Interaction: Lys235 (Donor–Donor), Leu236 (Acceptor–Acceptor)
Ginsenoside Rh2	Tunnel site	−8.8	Conventional Hydrogen Bond: Thr218, Arg229, Glu249, Thr253, His114, Arg111, Glu110 Van der Waals Interactions: Leu125, Glu232, Glu225, Lys129, Thr219, Glu250 Alkyl Interaction: Lys129

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
