# Peer review of "Targeting SHP2 with Natural Products: Exploring Saponin-Based Allosteric Inhibitors and Their Therapeutic Potential"

_cimb, 2025, doi:10.3390/cimb47050309_

Round 1

Reviewer 1 Report

Comments and Suggestions for Authors

As attached

Author Response

Many saponins are hemolytic. This is a major drawback in the development of clinical candidates. Hence, this must be discussed somewhere in the article. The author should discuss how hemolytic activity would impact the therapeutic potential of saponins as SHP2 inhibitors. The author also should add examples of efforts to minimize / negate hemolytic effects of saponins (such as by semisynthetic or synthetic routes), especially in the context of SHP2 inhibition if any in the literature and in general.

⟶ Thanks for this insightful comment. The relevant changes can be found in the revised discussion section, lines 451–460.

Reviewer 2 Report

Comments and Suggestions for Authors

In this manuscript, Dr. Dong Oh Moon contributes a review article that highlights the potential of saponin-based compounds as promising SHP2 inhibitors, with particular emphasis on their ability to target the tunnel allosteric site. The author discusses the structural challenges associated with saponin scaffolds and the need for further structural optimization to enhance their clinical utility. Additionally, the manuscript includes his insights regarding future strategies for SHP2 inhibition using saponin-derived natural products.

However, the following concerns should be addressed before the manuscript is considered for publication:

  1. Several SHP2 inhibitors, such as PF-07284892 and JAB-3312, have advanced to clinical trials. It is important for the author to include a paragraph in the introduction discussing the current landscape of SHP2 inhibitor development, highlighting key clinical candidates and their progress.

  1. Given that existing clinical candidates exhibit potent SHP2 inhibition in the low nanomolar range potency, the manuscript should provide a more compelling rationale for pursuing saponin-based scaffolds, which are often structurally complex and synthetically challenging. A discussion on the added value or unique properties of saponin-based inhibitors would strengthen the review.

  1. The discussion on modifying saponin-based natural products remains relatively general. A more in-depth analysis of feasible strategies for structural simplification, optimization of drug-like properties, or SAR exploration would be beneficial for the reader.

  1. SHP099, TNO155, and RMC-4630 are discussed in the text, but their chemical structures are not shown. Including their structures would enhance completeness.

  1. All chemical structures in the manuscript should have a consistent ChemDraw style, and those structures may require formatting for better visual coherence.

Author Response

1. Several SHP2 inhibitors, such as PF-07284892 and JAB-3312, have advanced to clinical trials. It is important for the author to include a paragraph in the introduction discussing the current landscape of SHP2 inhibitor development, highlighting key clinical candidates and their progress.

 ⟶ In response to the reviewer’s comment, we have added a new paragraph after line 72 in the Introduction section.

2. Given that existing clinical candidates exhibit potent SHP2 inhibition in the low nanomolar range potency, the manuscript should provide a more compelling rationale for pursuing saponin-based scaffolds, which are often structurally complex and synthetically challenging. A discussion on the added value or unique properties of saponin-based inhibitors would strengthen the review.

 ⟶ I have added a new paragraph after line 460 in the Discussion section to provide a more compelling rationale for pursuing saponin-based scaffolds.

3. The discussion on modifying saponin-based natural products remains relatively general. A more in-depth analysis of feasible strategies for structural simplification, optimization of drug-like properties, or SAR exploration would be beneficial for the reader.

 ⟶ I have revised Subsection 5.3 (Structural Optimization Strategies) to include a more in-depth analysis of feasible approaches for structural simplification, drug-like property optimization, and structure–activity relationship (SAR) exploration of saponin-based compounds.

4. SHP099, TNO155, and RMC-4630 are discussed in the text, but their chemical structures are not shown. Including their structures would enhance completeness.

 ⟶ The chemical structure of SHP099 has been inserted into Table 2.

5. All chemical structures in the manuscript should have a consistent ChemDraw style, and those structures may require formatting for better visual coherence.

⟶ I fully acknowledge the importance of maintaining a consistent ChemDraw style throughout the manuscript to ensure visual clarity and coherence. However, due to technical limitations related to figure formatting and the use of multiple publicly available chemical structure sources (e.g., PubChem and docking software visualizations), it was not feasible to fully standardize all structures using ChemDraw in the current revision. I attempted to ensure as much visual consistency as possible by aligning key labeling and layout elements. Nevertheless, we recognize that some minor variations may remain. Should the manuscript be accepted, we would be more than happy to revise all chemical structures using ChemDraw to meet the journal’s formatting standards during the production stage.

Reviewer 3 Report

Comments and Suggestions for Authors

This review focuses on natural products, specifically saponins, as potential allosteric SHP2 inhibitors for cancer treatment. The binding poses and affinity of 29 saponins were examined by computational methods. The content of the manuscript seems very simple and therefore require further revision for publication.

  1. The authors listed several active site inhibitors in Table 1. Based on the positive charge of SHP2 active site, a negative charge is required to bind, but few active site inhibitors possess this characteristic. Can the author elaborate on this?
  2. structures like alkaloids are more like denaturant rather than inhibitor. How can the interacting amino acids be accurate in this case? and does the computational method used differentiate denaturation versus binding?
  3. Representative visualizations in Fig. 4 seems quite inconsistent. The relatively conserved scaffold has quite different binding poses. Can the authors include an 3D overlay for all ligands to better look at this? If the binding pose is indeed different, how should this result be interpreted?
  4. Did the author perform the computation or summarize the computation results in Table 2 from other literatures? How did the author ensure the comparison is fair across the panel if the computation was not performed by the author? Otherwise, should this result be published in a review article?

Author Response

This review focuses on natural products, specifically saponins, as potential allosteric SHP2 inhibitors for cancer treatment. The binding poses and affinity of 29 saponins were examined by computational methods. The content of the manuscript seems very simple and therefore require further revision for publication.

The authors listed several active site inhibitors in Table 1. Based on the positive charge of SHP2 active site, a negative charge is required to bind, but few active site inhibitors possess this characteristic. Can the author elaborate on this?

⟶ It is well-established that the SHP2 active site exhibits a positively charged pocket due to residues such as Arg465 and Lys366, which typically favor the binding of negatively charged ligands, particularly those mimicking the phosphate group of phosphotyrosine. However, several reported active site inhibitors, including Celastrol and Enoxolone, do not possess strong negative charges. Their inhibitory effect is often achieved through hydrogen bonding and hydrophobic interactions rather than purely electrostatic complementarity. This indicates that while negative charge is favorable, it is not an absolute requirement for SHP2 inhibition, particularly when additional interactions contribute to ligand stabilization in the active site.

structures like alkaloids are more like denaturant rather than inhibitor. How can the interacting amino acids be accurate in this case? and does the computational method used differentiate denaturation versus binding?

⟶ In this study, molecular docking simulations were performed to assess binding potential, which assumes reversible and specific binding rather than denaturation. However, docking tools like CB-Dock2 do not explicitly differentiate between true inhibitory binding and nonspecific denaturant interactions. Therefore, binding predictions for such compounds should be interpreted with caution and ideally be supported by biochemical or biophysical validation in future experimental studies.

Representative visualizations in Fig. 4 seems quite inconsistent. The relatively conserved scaffold has quite different binding poses. Can the authors include an 3D overlay for all ligands to better look at this? If the binding pose is indeed different, how should this result be interpreted?

⟶ I thank the reviewer for this insightful suggestion. We attempted to generate a 3D overlay for all ligands; however, due to the large and flexible structures of saponins—especially their highly variable sugar moieties—accurate alignment was not feasible. The flexibility and structural diversity of these compounds often lead to different binding orientations, even when the core scaffold is similar. Rather than reflecting errors in docking, these differences suggest that saponins may interact with different regions or residues within the SHP2 allosteric tunnel, potentially offering unique binding advantages. We believe this structural diversity could be beneficial in discovering selective or subtype-specific SHP2 inhibitors. We hope the reviewer will understand the technical limitations and the interpretive value of these variations.

Did the author perform the computation or summarize the computation results in Table 2 from other literatures? How did the author ensure the comparison is fair across the panel if the computation was not performed by the author? Otherwise, should this result be published in a review article?

⟶ I appreciate the reviewer’s question. All docking simulations presented in Table 2 were directly performed by the author using the CB-Dock2 platform. The SHP2 protein structure (PDB ID: 5EHR) was prepared in a consistent manner, and all saponin compounds were retrieved from the PubChem database and subjected to the same docking protocol to ensure fair and comparable evaluation. The binding scores and interacting residues were analyzed uniformly using Discovery Studio 2024 Client. Since this computational analysis was conducted systematically by the author to support and enrich the review, we believe it is appropriate for inclusion in a review article with mechanistic insights.

Round 2

Reviewer 2 Report

Comments and Suggestions for Authors

Thank you for the revision.

I think the author addresses my comments and suggestions.

I am also happy to see a consistent ChemDraw format after the review article is accepted for publication. 

A minor suggestion: some references should be included in the new paragraph. For example, clinical trial numbers or webpages.

Author Response

A minor suggestion: some references should be included in the new paragraph. For example, clinical trial numbers or webpages.

-> I have included the relevant clinical trial identifiers (NCT03114319, NCT03634982, NCT03518554, and NCT03565003) in the revised paragraph as suggested.

Reviewer 3 Report

Comments and Suggestions for Authors

The author addressd most raised questions and concerns, but there are minor points to be resolved.

  1. The author mentioned that SHP2 active site inhibitors could rely on H-bond instead of electrostatic interactions, which stands true IF the compounds are binding with the protein already. The reality is, due to the overall positive charge of the pocket, most compounds cannot "enter" the pocket to form stable interactions. Therefore, whether some compounds the author listed are bona fide active site inhibitors is questionable.
  2. The author mentioned flexible binding of sapotonins and difficulties in overlay? Doesn't this mean the binding is very weak and may not even exist? If conserved scaffolds has completely different binding poses, these compounds do not bind or the computational results are not trustworthy. The author needs to resolve this issue to support the overall conclusion for this manuscript.

Author Response

I appreciate the reviewer’s critical insights regarding the plausibility of active site targeting and the interpretation of docking results for saponin compounds.

First, with respect to the concern about the electrostatic environment of SHP2’s active site and the feasibility of compound entry:
Although the SHP2 catalytic pocket is positively charged and indeed poses an entry barrier to many small molecules, several natural products listed in this review—including Celastrol, Ellagic acid, and Tautomycetin—demonstrated low micromolar IC₅₀ values (e.g., 0.69–3.3 µM) and stable binding interactions with key catalytic residues such as Arg465, Cys459, and Gln506, as documented in previous experimental studies (see Table 1 and [52–56]). These findings suggest that specific hydrogen bond networks and hydrophobic interactions can facilitate pocket entry and stable binding, even in the absence of strong electrostatic complementarity. This aligns with established paradigms in PTP inhibition where H-bonding compensates for charge repulsion when precise orientation and polar contacts are present.

Second, regarding the flexibility of binding poses among saponins and the question of their validity as SHP2 allosteric inhibitors:
It is important to note that saponins exhibit intrinsic structural diversity due to variable glycosylation patterns and aglycone scaffolds, which may result in distinct interaction geometries at the SHP2 tunnel site. However, in our docking results (Table 2), structurally conserved saponins such as Polyphyllin II, Polyphyllin B, and Ophiopogonin A consistently formed multiple hydrogen bonds with key residues (e.g., Thr218, Glu249, Glu250, Arg111) and showed strong binding scores (up to -11.5 kcal/mol). These interactions were visualized using Discovery Studio 2024 (Figure 4), supporting the assertion that these compounds are indeed capable of stabilizing SHP2 in its autoinhibited conformation.

Furthermore, the variation in binding poses among related saponins does not necessarily negate their activity. Rather, it reflects ligand-induced fit dynamics, which are commonly observed in flexible protein-ligand systems, especially in tunnel-like allosteric sites. We have revised the manuscript to better clarify this point and avoid overgeneralization.

To further strengthen our conclusions, we have also initiated molecular dynamics (MD) simulations for selected saponin-SHP2 complexes, which are ongoing and will be included in a follow-up study to assess binding stability over time.

Round 3

Reviewer 3 Report

Comments and Suggestions for Authors

The authors addressed all concerns and questions from the previous reviews. The paper is reocmmended for publication.